# TableVault: Managing Dynamic Data Collections for LLM-Augmented Workflows

Jinjin Zhao
j2zhao@uchicago.edu
University of Chicago
Chicago, IL, USA

Sanjay Krishnan
skr@uchicago.edu
University of Chicago
Chicago, IL, USA

## Abstract

Large Language Models (LLMs) have emerged as powerful tools for automating and executing complex data tasks. However, their integration into more complex data workflows introduces significant management challenges. In response, we present TableVault - a data management system designed to handle dynamic data collections in LLM-augmented environments. TableVault meets the demands of these workflows by supporting concurrent execution, ensuring reproducibility, maintaining robust data versioning, and enabling composable workflow design. By merging established database methodologies with emerging LLM-driven requirements, TableVault offers a transparent platform that efficiently manages both structured data and associated data artifacts.

## CCS Concepts

• **Theory of computation** → *Data provenance*; • **Information systems** → **Storage management**; • **Computing methodologies** → **Information extraction**; **Intelligent agents**; *Knowledge representation and reasoning*.

## Keywords

LLM Agents, ETL Systems, Provenance, Dataframes, Document Retrieval

**ACM Reference Format:**
Jinjin Zhao and Sanjay Krishnan. 2025. TableVault: Managing Dynamic Data Collections for LLM-Augmented Workflows. In *Proceedings of Novel Optimizations for Visionary AI Systems Workshop @SIGMOD (NOVAS '25).* ACM, New York, NY, USA, 5 pages. https://doi.org/3735079.3735321

## 1 Introduction

Large Language Models (LLMs) significantly advance task automation but are often inefficient alone, requiring integration into larger data workflows [2, 3, 24, 29, 32]. In these workflows, LLMs generate or transform data artifacts for analytical pipelines, such as by creating report versions or structured summaries for downstream queries. This evolution presents a fundamental challenge: efficiently managing interconnected workflows and data artifacts, rather than isolated processes. In these complex systems, tracking data lineage, versioning transformations, and managing concurrent operations becomes crucial. Several critical needs emerge:

**(1) Diverse Data Transformation:** Effectively managing lineage for varied transformation patterns [21, 33] to enable parallel processing and other performance benefits.

**(2) Operation Safety and Reproducibility:** Ensuring operations execute safely without corrupting system state, and that data artifacts are reproducible for accuracy in automated workflows.

**(3) Data Versioning:** Robust mechanisms for versioning dynamic data (due to model evolution, data drift, updates) to support incremental improvements, audits, and error recovery.

**(4) Workflow Composability and Transparency:** Making all generated artifacts readily accessible for downstream tasks to foster collaboration and enable data repurposing for new applications.

To address these challenges, we present **TableVault**, a unified platform integrating data management, table generation, and LLM execution. TableVault merges traditional database principles with modern workflow management, explicitly tracking operations, lineage, and performance for transparent and reproducible complex data workflows. It treats data artifacts and workflows as an interconnected repository with human-readable and machine-accessible records, enhancing governance and easing integration with external systems. Ultimately, TableVault is designed to streamline dynamic data collection orchestration, improving workflow efficiency, operational safety, and overall system complexity management.

In the following sections, we detail the architectural components, operational methods, and execution frameworks that underpin TableVault. Through its design, TableVault aims to improve the management of LLM-augmented workflows, enabling future opportunities for enhanced performance and accuracy.

## 2 System Outline

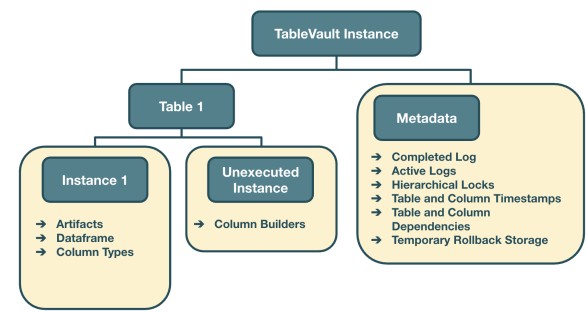

**Figure 1: System diagram of TableVault.**

To manage evolving data artifacts and dataframes, TableVault employs a specific storage structure (as shown Figure 1). Table objects are stored as folders within a TableVault directory alongside a folder containing metadata files. Each table object is designed to serve a clearly defined conceptual purpose while containing versioned instances that may evolve over time. This file-based system, with directly accessible, human-readable files, promotes governance and system state transparency.

Table instances are versioned by their explicit generation timestamp, optionally with a user-defined identifier. An instance includes user-defined YAML specification files, called "builders," that describe how to form columnar data within the table. Builders specify explicit parameterization for Python functions and language model APIs and serve as a transparent record of the provenance of the linked dataframe. Many language model execution systems similarly use user parameterization to execute APIs[12, 24]. In TableVault, builders can also track data lineage by allowing references to other TableVault tables, thereby enabling more composable and complex data workflows. Builder parameters can be easily adjusted between instances before materialization; for example, users may reword a language model message to increase accuracy or modify the underlying model to improve performance.

Upon generation, a dataframe object and optional artifacts -such as documents or images - are created and stored in the table instance. Within a table, either all generated instances and associated artifact collections can be active and referenced, or only one instance (or none) may be active at a time. The latter approach may be preferable if the table is linked to external side effects. Section 4 describes builders and execution in detail.

The TableVault instance's `metadata` folder is crucial for robust workflow orchestration and addressing operational challenges. It stores: 1) metadata regarding dataframe and column versions, including dependencies vital for lineage tracking; 2) a centralized log of all active and completed system operations, ensuring auditability; and 3) hierarchical shared and exclusive locking mechanisms at both table and instance levels. This locking ensures data consistency during concurrent operations — a key aspect of managing complex data systems — as locking a table implicitly locks all its instances.

## 3 TableVault Operations

### 3.1 Standard Operations

The TableVault system supports various standard operations that execute in background threads, allowing a single-threaded application (e.g., Jupyter Notebooks) to run multiple operations concurrently. Every execution is assigned a unique identifier that can be used to stop or restart it. This capability is particularly useful for early stopping of table instances; for example, if an error in a language model message is detected before it is applied to every row of a dataframe, API costs can be reduced. Additionally, each execution is linked to a unique system process ID to prevent multiple active processes for the same operation. Some significant operations include:

**Create Table.** This operation creates a new table within a TableVault instance. The behavior of its instances is specified — for example, whether multiple active instances are allowed and whether new instances can run concurrently with an ongoing execution.

For instance, if multiple instances iterate over different language model messages, several active instances may be relevant; however, if a table is linked to file identifiers uploaded to OpenAI, earlier instances may become outdated when a new execution begins.

**Create Table Instance *and* Generate Table Instance.** This operation creates a new table instance that has not yet been generated. Builders from previous versions of the table can be copied into this new instance, allowing users to adjust them before generating the instance. Once generated, the instance is made accessible to other tables.

**Delete Table *and* Delete Table Instance.** This operation deletes the dataframe and artifacts in a table or table instance. Although ideally every instance of every table would be retained, storage constraints often make this infeasible. Since the YAML builders and execution times are maintained, all tables could be recovered if the external environment is preserved (aside from natural output variability in language model responses).

The full list of operations is summarized in Table 1. Every operation that writes to a table is logged along with the user who executed it, aiding in the identification of malicious activities. Metadata is updated concurrently with these operations to ensure that the state of the system remains consistent.

Figure 2 illustrates an example of creating and executing a TableVault instance using the library's Python interface. In this example, a TableVault table called "`stories`" is created. The "`gen_stories`" builder file is copied from the "`..\test_data...`" directory into the table, and an instance is created and executed from that builder.

```python
tablevault = TableVault(db_dir="test", author="jinjin", create=True)
tablevault.setup_table('short_stories', allow_multiple_artifacts = False)
tablevault.copy_files("../test_data/stories", table_name="short_stories")
tablevault.setup_temp_instance("short_stories", builder_names=["gen_stories"])
```

**Figure 2: Example Python Execution of TableVault Operators.**

**Table 1: TableVault operations and their descriptions.**

| Example | Description |
|---|---|
| Create TableVault Instance | Create a TableVault instance. |
| Create Table | Create a new table in a TableVault instance. |
| Create Table Instance | Create a new (unexecuted) table instance. |
| Copy Builders | Copy builders from an external file location into a table. |
| Generate Table Instance | Execute a table instance. |
| Delete Table | Delete a table. |
| Delete Table Instance | Delete a table instance. |
| Restart TableVault Instance | Restart a TableVault instance and resume stopped processes. |
| Get Active Processes | Retrieve currently active processes. |
| Get DataFrame | Retrieve the dataframe from a table instance. |

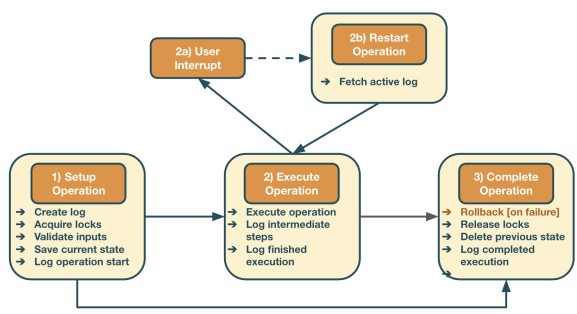

**Figure 3: Execution flow of write operations.**

## 3.2 Write Operation Execution

Figure 3 details TableVault's write operation execution flow. Each operation begins by creating an active log with a unique identifier. The system then saves the current state, validates inputs, and acquires necessary locks. Once locks are secured, inputs are logged, and execution commences. Critically, external interrupts during execution do not release acquired locks, allowing operations to be restarted; users can resume by re-executing with the same identifier. Upon successful completion, the previous state is deleted, locks are released, and the operation is marked complete. Internal failures or explicit stops trigger a rollback, reverting the system to its prior state.

This execution flow resembles traditional database operations but emphasizes user control over interrupts and restarts. This is crucial because TableVault executions (i.e., table instance generations) are typically longer and more resource-intensive than standard SQL operations. We ensure ACID properties (Atomicity, Consistency, Isolation, Durability) [8, 9, 15] using rollbacks, two-phase locking, and write-ahead logs. These measures guarantee that operations are safely monitored and maintain system integrity. As data workflows increasingly become autonomic [13] with reduced human intervention, designing systems with comprehensive operational logging and robust safeguards is vital for correctness and reliability.

## 4 Instance Generation

### 4.1 Builders

User-defined YAML files, named builders, direct dataframe creation for table instances by specifying column execution and program definition parameters. A builder can produce multiple columns, but each column originates from one builder. Builders can implicitly define common data transformation patterns using the TableString reference. Defining these parameters explicitly creates opportunities for execution optimization. Explicit parameter definition facilitates execution optimizations: basic ones apply by default in TableVault, while complex strategies are potential system extensions or future research [13, 24]. Multiple types of builders can be declared in the TableVault system.

Table 2 shows the property specifications for an example builder type, *Code Builder*. The *Code builder* executes a Python function that generations dataframe columns or rows. All builder specifications require generated column names, the number of allocated threads, and (optional) column datatypes. This demonstrates how

builder specifications defines both program parameters and execution conditions of column generation. LLM API calls can be made directly with Code builders or via specialized wrapper builders, like our *OpenAI Thread builder* for the OpenAI API [18] (not shown).

TableVault allows creating new builder types by defining their parameters and execution strategy. Its flexible model supports exploring execution strategies without building a new platform [24].

**Generator Builder.** Each table instance requires a table generator builder, identified by a 'gen_' YAML prefix. This builder, of any type, creates row record identifiers (defining dataframe entries). Current implementations include one using external artifact directories and another using existing TableVault tables.

**Artifacts.** Artifacts are simply collections of files that are not well-suited to be stored within a dataframe. In language model workflows, artifacts can be documents or images provided to or generated by the language model. For system consistency, artifacts are not directly referenced; rather, they are fetched using the linked dataframe object. To store artifacts within TableVault, we simply move the files to the destinated artifact folder and define a column in the dataframe with the `artifactartifact` datatype. Generated artifacts can be retrieved in any builder specification by referencing the linked dataframe column with a `TableString`.

**Table 2: Builder properties and their descriptions.**

| Property | Description |
|---|---|
| Changed Columns | Column names to be generated (e.g., "[question-1, question-2]"). |
| N-Threads | Number of threads. |
| Column Datatype | Datatype of columns (optional). Allows the "artifact" datatype. |
| **Code Builder** | |
| Row-Wise | If true, generates a single row; if false, generates an entire column. |
| Python Function | Module and function name. |
| Arguments | Function arguments (e.g., "{ df: «financial_docs» }"). |
| Row-Save | If true, save after each row; if false, save after each builder. |

### 4.2 TableString and Data Processing Patterns

A `TableString` reference serves as a dynamic placeholder within a prompt file, which is substituted at runtime with a dataframe, vector, or constant value from a table instance within the same TableVault. This mechanism allows for flexible data referencing inbuilders. The structure of a `TableString` is illustrated in Figure **??**[cite: 107, 137]. It requires a table name and optionally accepts a specific instance identifier (defaulting to the latest instance at execution time), a column selection, and row filters. The row filter functions akin to the `df.query` method in the pandas library, enabling selection based on a column key and a value or range[19].

Reserved keywords enhance the utility of `TableString` references. The keyword `SELF` refers to the partial dataframe currently being generated, while `INDEX` denotes the dataframe's row index. In row-wise prompt executions, `SELF.INDEX` provides a concise

way to access the index of the row being processed. Crucially, `TableString` references facilitate the tracking of data provenance by explicitly defining dependencies between prior table instances and the current one being generated.

Furthermore, the row filtering capability within `TableString` enables the representation of common data transformation patterns [21, 33]. These patterns can be applied to tasks such as aggregating paragraph artifacts into a document or executing Retrieval-Augmented Generation (RAG) queries across a collection of documents in a table. Examples of how this is achieved are shown in Table 3. Providing no indices to a column-generating Code function results in a reduce transformation (1). Matching the dependency table instance's index with the current table instance's index performs a one-to-one transformation (2). Using a range from the first index to the current row index achieves an aggregation (3). Employing relative ranges around the current index facilitates a convolution (4). Selecting rows based on specific column values implements an equality selection (5).

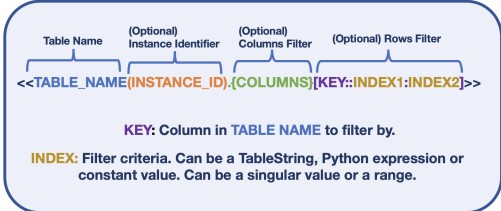

**Figure 4: Breakdown of TableString components.**

**Table 3: Example TableString indices for a row-wise Python function argument.**

|   | Example | Pattern |
|---|---|---|
| 1 | *NONE* | Reduce |
| 2 | INDEX :: SELF.INDEX | One-to-One |
| 3 | INDEX :: 0 : SELF.INDEX | Aggregation |
| 4 | INDEX :: SELF.INDEX - 5 : SELF.INDEX | Convolution |
| 5 | FRUIT_COLUMN :: "apples" | Selection |

### 4.3 Execution Flow and Optimizations

We now describe the execution flow for generating a dataframe from builders. Specific optimizations are implemented to prevent unnecessary recomputation, which is important because computing entries can be expensive (e.g., a single entry might require multiple LLM API calls).

Figure 5 illustrates the steps of this execution. First, TableVault extracts table dependencies from the builders (not shown). Next, we assess updates in the table dependencies and builder files relative to the previous execution of the same table. Columns in the dataframe for which the dependencies and builders remain unchanged are retained from the previous instance (1). Then, the generator builder is executed, and its output is left-joined with the current dataframe, ensuring that only new rows have empty entries (2). Subsequently,

builders are executed only on the empty entries of the dataframe (3). If there are dependencies among the builders (captured by the "SELF" keyword), they are executed in the order specified by the topological tree. Multiple threads can be used for execution if specified.

The execution flow can be safely started and stopped, as discussed in the previous section, and it resumes from the last disk write of the table. The frequency of disk writes depends on the cost of the operation (i.e., since LLM calls are inherently more expensive, they should be saved more frequently).

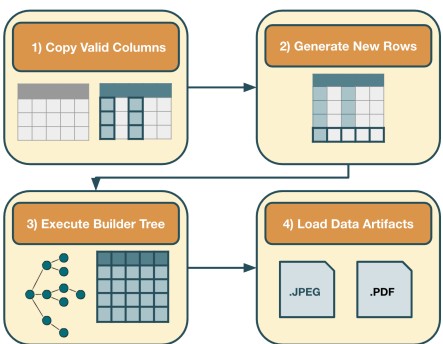

**Figure 5: Generation flow of table instances.**

## 5 Applications and Related Works

**Workflow Optimizations.** Prior research optimized prompt pipelines for performance and accuracy [12, 13, 20, 24, 27] through methods like algorithmic model selection, operation reordering, cost estimation via data sampling, and prompt caching. As TableVault can implement these as program-generated prompts, it may enable faster, more customized pipeline optimizations.

**Complicated RAG and Document Analysis Patterns.** Beyond single-document Retrieval-Augmented Generation (RAG), complex applications include language models augmented with knowledge graphs detailing inter-artifact relationships [3, 7, 10, 11, 22, 25]. These can enhance product recommendations [31] and identify concept similarities in scientific research [3]; language models can also generate these graphs. Another complex pattern uses language models for data clustering and labeling [2, 16, 28, 32]. TableVault is designed to support reliable workflow execution and tracking in these scenarios. System challenges with dynamic or changing data, like execution scheduling, materialization, and data drift, are well-suited for TableVault.

**Anomaly Detection.** In systems like TableVault, where machine agents perform data modifications [23], detecting anomalous behavior is vital. Research addresses anomaly detection in data pipelines and operational logs [4, 6, 14, 30], and applying these to agent-based systems is pertinent.

**Comparison to Other Systems.** TableVault shares commonalities with traditional databases and ETL systems [1, 17, 26] requiring data operation durability and reliability. TableVault is an evolution, enabling complex data transformations, language model task execution, and data provenance tracking. While similar to language model prompting tools [5, 12], TableVault uniquely emphasizes data lineage and cross-workflow execution optimizations.

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
