# OpenReview forum: "TableVault: Managing Dynamic Data Collections for LLM-Augmented Workflows"
_ACM.org/SIGMOD/2025/Workshop/NOVAS — NOVAS 2025_

### Official Review · Reviewer_5isV · 2025-04-16

**Confidence:** 4

**Improvement Opportunities:**

- The paper often dives into low-level details early on, making it difficult for the reader to follow the high-level ideas. A better flow would be to begin with a concrete end-to-end example and then gradually explain the underlying components. For example, move Figure 5 earlier.
- There are no experimental evaluations. The paper should clearly define metrics to measure utility, overhead, and usability, even without having an evaluation and baselines to compare them with.
- It would be helpful to discuss the potential overhead of maintaining the TableVault structure. In which cases do we see diminishing returns or unnecessary complexity?

**Minor Comments:**

See my comments in "Improvement Opportunities."

**Short Summary:**

TableVault is a unified platform for language model–augmented workflows that supports detailed tracking of data operations, workflow execution patterns, and data provenance. It integrates data management, table generation, and LLM execution into a system that combines traditional database principles with workflow management.

**Strong Points:**

- Tackles an important and timely question: How can we efficiently manage and orchestrate a repository of interconnected workflows and data artifacts, rather than treating each workflow in isolation?

- The paper is clearly written, with a clean separation between system design and operational details.

- The system implementation is described in detail, demonstrating the authors' careful engineering effort. However, in some places, the level of detail may overwhelm the reader.

---

### Official Review · Reviewer_Gvrg · 2025-04-18

**Confidence:** 4

**Improvement Opportunities:**

W1. Data model is not clear. Are tables the primarily supported data model in TableVault? TableVault claims a lot of functionalities, such as logging and versioning. However, a general workflow system does not restrict operations only to tables. It can be data processing on structured or unstructured data, such as document analytics. The paper mentions RAG as one possible application for TableVault, but I don’t see why, since RAG does not necessarily happen on tabular data. It can take a document as input, apply a filter operator to identify a document subportion, then forward it to LLMs to answer the questions. In such a simple workflow, how TableVault works in this setting is not clear. It also mentions that other data modalities, such as documents or images, can be stored in a table instance. How can this be achieved? An example for the data model, and how the model can support different types of data in the system, is appreciated.

W2. Lineage Maintainence: Let’s assume TableVault focuses on table operations. If that is the case, maintaining the lineage among table instances for purposes such as provenance seems trivial. We can always maintain the relationships along the data flow in a workflow system. What is the key insight here?

W3. Consistency - Write operation flow. TableVault seems to put a control on the write flow to ensure consistency and claims the user’s loop in the flow is the key difference with traditional DBs. If that’s the case, how can you ensure that the user’s flow will not break the consistency implemented by standard protocols, such as 2-phase locking?

W4. Data processing patterns are vague. What are the data processing patterns referred to in the paper? This is not clear, so I have trouble understanding the other motivations related to this, such as how to capture that, how to leverage that, and for what purpose?

W5. How TableVault benefits workflow optimizations is not clear. To store what logs, and how, can benefit what optimizations? The paper claims too many functionalities, but few of them are clear in the paper. It would be great if you can put some insightful examples per aspect to illustrate why it is interesting, and at a high level, how TableVault can help.

**Minor Comments:**

A high-level envisioning of the system is OK, but bringing out the truly insightful ideas about how the system can support and improve upon the mentioned functions is appreciated.

**Short Summary:**

This paper presents TableVault, a system that manages workflows, providing support for versioning, consistency, optimization, and lineage.

**Strong Points:**

S1: Clear presentation.

S2: TableVault envisions many functionalities, many of which are interesting in this space.

S3: The paper provides applicable applications and some examples to illustrate the proposed system, which helps with understanding.

---

### Official Review · Reviewer_RyGY · 2025-04-25

**Confidence:** 3

**Improvement Opportunities:**

(W1) The motivation for the system is LLM-augmented workflows, but the resulting system seems to be designed to support any data science workflow.

If we observe the list of emerging challenges that the authors identify, we can see that none of them are intricately tied to the emergence of LLMs. They seem to be the same challenges that naturally arise in almost all data science and advanced analytics workflows with lots of artifacts and long-running jobs. Hence, even though I believe those challenges are valid, the way the authors frame their system design is like it is geared specifically towards LLM-augmented workflows. However, I wasn't able to find any LLM-specific challenges or any design choices that are specifically only for this setting.

(W2) Given that the system resembles a general data analytics system for long-running workloads, the authors ought to compare with and place their contributions in the context of related work in that area.

I understand the enthusiasm for LLMs, but I believe it is important to acknowledge that their emergence didn't exactly erase the previous progress made in the data management field and the systems for data science and advanced analytics. Hence, the authors should place their work in the context of the related systems that have been developed over the years in order for the reader to get a fair impression of the novelty of the proposed system.

(W3) Some of the terminology is slightly confusing.

For example, the YAML files that describe how a table is generated are called "prompts". I understand that the intention of the term was to connect it to LLMs, but the connection feels a bit forced and slightly confusing. Also, as far as I understood, the term "instance" seems to be used for two separate things: (1) the instance of a table vault, and (2) a version of a table. As an example of the confusing usage of the word instance, we can look at the following sentence:

"For instance, if multiple instances iterate over different language model messages, several active instances may be relevant; however,
if a table is linked to file identifiers uploaded to OpenAI, earlier instances may become outdated when a new execution begins."

This terminology feels slightly confusing to me and should probably be cleared up.

**Minor Comments:**

n/a

**Short Summary:**

The authors present the main aspects of their system for managing data in LLM-augmented analytical workflows. They start off by making the observation that in modern data processing pipelines, LLMs are responsible for generating and/or processing many data artifacts. As a result, they highlight several emerging challenges: (1) the presence of a variety of transformational patterns with an increasing amount of them requiring long runtimes; (2) the need for safe execution that does not jeopardize the state of the system, coupled by reproducible results that enable the recreation of artifacts in case of loss; (3) dynamically changing datasets that require versioning support; and (4) workflows that are easily composable to perform more complex tasks, while at the same time transparent, making them easy to inspect. The authors adopt the relational model as the data model of choice and structure a "table vault" as a collection of tables, each one represented by one or more versions of data (i.e. instances) along with table-specific metadata that specifies how the table is generated (referred to as a "prompt"). Apart from tabular data, the tables can store other data artifacts such as images or documents. Tables are linked by data transformations, forming a lineage graph. The system is in charge of executing the transformations of the lineage graph, balancing efficiency with ACID properties.

**Strong Points:**

(S1) The authors make a solid argument that the emerging space of LLM-augmented workflows is in need of better data management systems in order to make them more reliable and efficient.

(S2) The authors leverage many good ideas from the field of data management that seem like overall pretty good design choices. This includes using the relational model as the core data model, tracking the lineage of all data artifacts, etc.

(S3) The system strives to keep all data artifacts and their associated metadata in human-readable file formats as much as possible, which to me seems like a good design choice because it will make the system feel more transparent to the user, and hence more trustworthy.